# The ITS analysis and identification of *Actinidia eriantha* and its related species

Xiaoqin Zhang[1,2‡], Yixing Zhang[3‡], Jiale Mao[1], Yan Lan[1], Zunjing Zhang[1]*, Houxing Lei[1]*

1 Department of Pharmacy, Lishui Hospital of Traditional Chinese Medicine, Lishui, Zhejiang, China, 2 School of Pharmacy, Beijing University of Traditional Chinese Medicine, Beijing, China, 3 Longhua Hospital Shanghai University of Traditional Chinese Medicine, Shanghai University of Traditional Chinese Medicine, Shanghai, China

‡ These authors contributed equally to this work and are considered as co-first authors.
* zjleihx@163.com (HL); zunjing123@163.com (ZZ)

**Data Availability Statement:** All relevant data are within the paper and its Supporting Information files.

## Abstract

The dried plant material of medically important plant *Actinidia eriantha* especially when it remains in the form of powder often look morphologically similar to its related species. The lack of efficient methods to distinguish the authentic material from other similar species leads to chances of adulteration. The molecular authentication of herbal plant materials such as the internal transcribed spacer (ITS) sequences is considered as more reliable method compared to morphological traits. In this study, we aim to evaluate the potential of identification for roots of *A. eriantha* and its related species by ITS sequences. The lengths of ITS regions ranged from 624 to 636 bp with GC content ranging from 50.96% to 59.55%. A total of 194 variation sites and 46 haplotypes were formed in 185 samples. Among them, the roots of *A. eriantha* possessed specific sites at 85bp (C), 205bp (T), 493bp (C), 542bp (G), 574bp (C), 582bp (T) and 610bp (G), while *A. hemsleyana*, *A. callosa*, *A. valvata* and *A. polygama* have their own specific sites. The inter-specific genetic distance among 8 *Actinidia* species in the range 2.28% to 11.00%. The phylogenetic tree constructed with ITS, ITS1 and ITS2 region showed that the ITS sequences have higher potential for identification in 8 *Actinidia* species. However, as to *A. eriantha*, *A. hemsleyana* and *A. valvata*, these three barcodes have the same identification ability. The ITS regions indicated that different samples from same species can be grouped together, except for *A. arguta* and *A. melanandrah*. In conclusion, the ITS sequences can be used as an efficient DNA barcode for the identification of *A. eriantha* and its related species.

## 1 Introduction

The quality and safety of medical materials are important guarantees for the health and legal rights of patients. The roots of *Actinidia Eriantha* Benth. is a traditional Chinese medicine. It has been included in the Standard of TCM processing in Zhejiang Province in 2015 [1], and has the function of clearing heat and detoxifying, dampness and swelling in the Traditional

**Funding:** Ms. Xiaoqin Zhang was funded by [Zhejiang Provincial Natural Science Foundation of China] grant number [LGF20H280005], Mr. Houxing Lei was funded by[development project in Lishui city] grant number [2020ZDYF 15] and Ms. Yan Lan was funded by [Lishui Science and Technology project] grant number [2021SJZC039]. The role of funders in research: Ms. Xiaoqin Zhang – Methodology and Writing–original draft. Mr. Houxing Lei –Conceptualization. Ms. Yan Lan – Software.

**Competing interests:** The authors have declared that no competing interests exist.

Chinese Medicines. The roots *A. Eriantha* are often used to treat breast carbuncle, rheumatic pain, tumor and other diseases [2, 3].

The genus *Actinidia* (family: Actinidiaceae) includes approximately 66 species, about 62 of which are distributed in China. Among them, *A. Eriantha*, *A. valvata*, *A. macrosperma*, and *A. arguta*, are identified as important medicinal species by the Standard of TCM processing in Zhejiang Province in 2015 [1]. However, it is difficult to identify them by leaves due to the diverse variability in morphological characters and short flowering and fruiting period [4]. And the medicinally important *A. Eriantha* species need exhaustive taxonomic expertise for taxonomic identification especially to discriminate among *A. Eriantha*, *A. valvata*, *A. macrosperma*, and *A. arguta*. The dried plant material of *A. Eriantha* in the form of powder looks very similar to *A. Eriantha*, *A. valvata*, *A. macrosperma*, and *A. arguta* and therefore lead to chances of adulteration [5]. For example, the root of *A. Eriantha* had good inhibitory effect on liver cancer and colorectal cancer [3, 6], whereas the *A. valvata* and *A. macrosperma* had obvious inhibitory effect on gastric cancer and lung cancer, and the anti-tumor effect of *A. arguta* was seldom reported [7, 8]. So, the adulteration in medicinal plants affect the efficacy of the drug Fig 1.

The DNA sequence based molecular authentication of herbal plant materials such as the internal transcribed spacer (ITS) sequences of nuclear ribosomal DNA (nrDNA) is considered as a more reliable method as compared to the methods based on morphological traits and biochemical methods [9, 10]. Although some scholars used SSR and RADP techniques to analyze the genetic diversity of the root of *A. Eriantha* [11, 12], and some scholars also studied the phylogenetic development of the genus of *Actinidia* by using *mat*K and *rbc*L sequences [13], no report hitherto has been found to use ITS sequences to conduct species identification for the root of *A. Eriantha* and its related species.

In this study, we attempted to use the ITS sequences for rapid identification of eight *Actinidia* species in China. This study will not only lay a foundation for phylogenetic analysis but also provides experimental and practical basis for the rapid authentication of *A. Eriantha*.

## 2 Materials and methods

### 2.1 Plant materials

All 8 *Actinidia* species were wild species, *A. Eriantha* were collected from Lishui city, Zhejiang Province and Nanping City, Fujian Province, in China, respectively, and other *Actinidia* species were collected from Lishui City, Zhejiang Province, in China. A total of 185 samples were collected, including 53 samples from Genbank gene database (Table 1). All voucher specimens are kept in Lishui Hospital of Traditional Chinese Medicine.

### 2.2 DNA extraction, amplification, and sequencing

Fresh young leaves of the collected samples were used for genomic DNA isolation, as previously reported. The pair of universal primers used to amplify nrDNA ITS sequences consist of P1 (5'– AGAAGTCGTAACAAGGTTTCCGTAGG– 3') and P4 (5'– TCCTCCGCTTATTGA-TATGC–3'). PCR amplification was conducted using 50 μL volumes containing 5×PCR Buffer 5 μL, dNTPs (5 mmol/L) 5 μL, MgCL (25 mmol/L) 4 μL and each primer (0.5 μM) 2 μL, and 1 U Taq DNA polymerase. PCR amplification was conducted with the following parameter settings: holding at 94°C for 5 min followed by 35 cycles at 94°C for 30 s, at 56°C for 45 s, and at 72°C for 1 min, and a final extension at 72°C for 10 min. The PCR products were analyzed by 1.0% agarose gel electrophoresis, and then were sequenced in Shanghai Shenggong Biotechnology Co., Ltd, Shanghai, China.

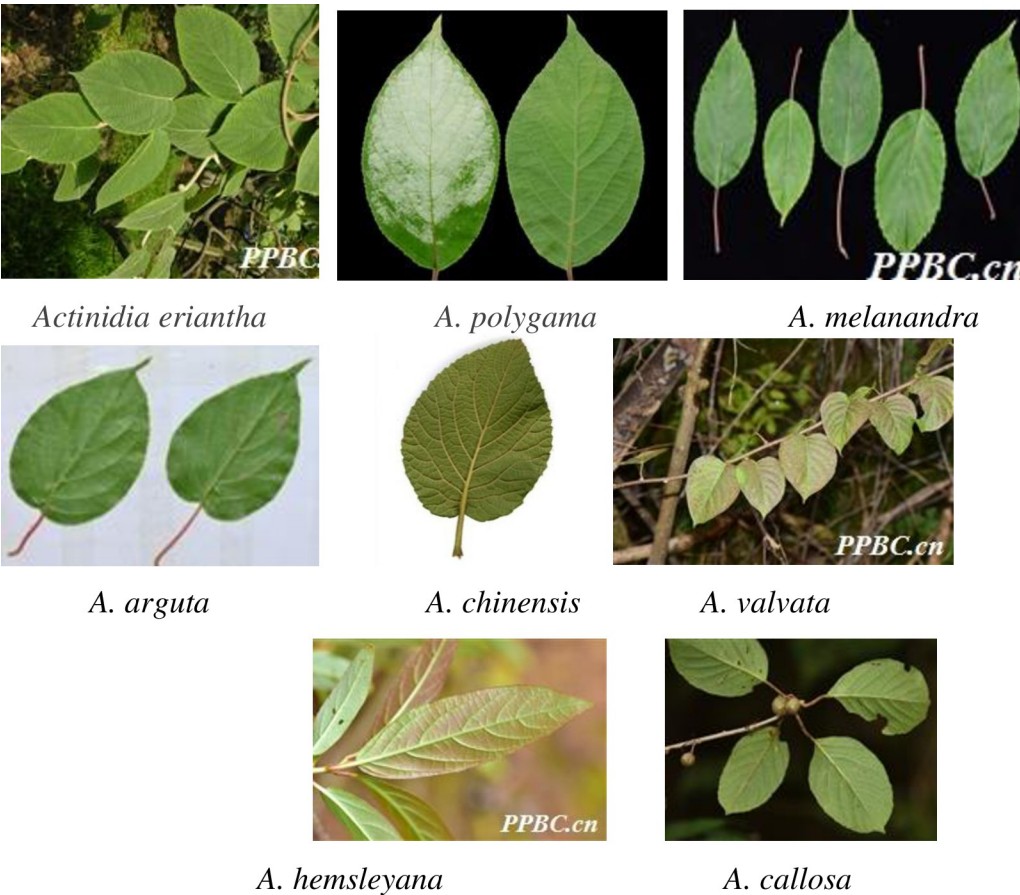

*Actinidia eriantha*　　　　*A. polygama*　　　　*A. melanandra*

*A. arguta*　　　　*A. chinensis*　　　　*A. valvata*

*A. hemsleyana*　　　　*A. callosa*

**Fig 1. Leaf image of 8 species of *Actinidia*.** Note: All pictures from Plant Photo Bank of China.

## 2.3 DNA analysis

The ContingExpress and DNAman software were used to align the ITS sequences, analyze the sequence length, variation sites, and this was assisted by manual editing. The Editseq and Mega-X software were used to construct the Phylogenetic tree, calculate the inter-specific and intras-pecific genetic distances. Phylogenetic analysis was conducted according to the maximum likelihood (ML), with the following parameter: 1000 bootstrap replicates. The inter-specific and intras-pecific genetic distances were conducted according to the K2-P models.

## 2.4 additional information regarding the permits

The research material is plant leaves, it is not involve ethical protection and genetic information protection, so no additional information regarding the permits for the work we need.

## 3 Results

### 3.1 PCR efficiency and characteristics of ITS sequences

The success rates of PCR amplification and sequencing of ITS regions from the sampled specimens were 100%. The lengths of ITS regions were ranged from 624 to 636 bp with GC content ranging from 50.96% to 59.55%. The ITS sequence was divided into ITS1, 5.8S and ITS2 sequences, as shown in Table 2. The 5.8s sequences of the eight *Actinidia* species were highly

**Table 1. Samples information.**

| Species Name | Locatiy information | Samples No. | Samples size | Haplotype(number) |
|---|---|---|---|---|
| *Actinidia eriantha* | Qingtian, Lishui, Zhejiang | DY(1~5) | 5 | MH1(4),MH2(1) |
| | Dagangtou,Lishui, Zhejiang | HK(1~5) | 5 | MH1(5) |
| | DAshanfeng,Lishui, Zhejiang | SF(1~12) | 12 | MH1(12) |
| | Wuyishan, Nanping, Fujian | WY(1~2) | 2 | MH1(1),MH3(1) |
| | Pucheng, Nanping, Fujian | PC(1~7) | 7 | MH1(4), MH3(2), MH4(1) |
| | Pucheng, Nanping, Fujian | FG | 1 | MH1(1) |
| | Longquan,Lishui, Zhejiang | AR(1~2) | 2 | MH1(2) |
| | Longquan,Lishui, Zhejiang | LQ | 1 | MH1(1) |
| | Shakengkou, Lishui, Zhejiang | SK(1~4) | 4 | MH1(2), MH3(1),MH5(1) |
| | Lishan,Lishui, Zhejiang | LL(1~13) | 13 | MH1(11), MH3(1), MH4(1) |
| | Genbank (AF323800) | Ae-1 | 1 | MH1(1) |
| | Genbank (MG714287) | Ae-2 | 1 | MH6(1) |
| | Genbank (MK425151) | Ae-3 | 1 | MH1(1) |
| | Genbank (AF323801) | Ae-4 | 1 | MH1(1) |
| | Genbank (KP314044) | Ae-5 | 1 | MH7(1) |
| | Genbank (KP314039) | Ae-6 | 1 | MH8(1) |
| | Genbank (KP314038) | Ae-7 | 1 | MH9(1) |
| *A. polygama* | Lishan,Lishui, Zhejiang | GT(1~10) | 10 | GZ1(1) |
| | Genbank (KP314060) | Ap-1 | 1 | GZ2(1) |
| | Genbank (KP314056) | Ap-2 | 1 | GZ1(1) |
| | Genbank (KP314051) | Ap-3 | 1 | GZ3(1) |
| | Genbank (KP314035) | Ap-4 | 1 | GZ1(1) |
| | Genbank (KC519766) | Ap-5 | 1 | GZ4(1) |
| | Genbank (KC519765) | Ap-6 | 1 | GZ4(1) |
| | Genbank (AF323796) | Ap-7 | 1 | GZ5(1) |
| | Genbank (MW234501) | Ap-8 | 1 | GZ2(1) |
| | Genbank (MW234500) | Ap-9 | 1 | GZ2(1) |
| *A. melanandra* | Lishan,Lishui, Zhejiang | HT(1~10) | 10 | HL1(1) |
| | Genbank (AF443211) | Am-1 | 1 | HL2(1) |
| | Genbank (MG714285) | Am-2 | 1 | HL1(1) |
| | Genbank (MG714282) | Am-3 | 1 | HL3(1) |
| | Genbank (AF323808) | Am-4 | 1 | HL4(1) |
| *A. arguta* | Lishan,Lishui, Zhejiang | RT(1~10) | 10 | WD1(1) |
| | Genbank (KP314061) | Aa-1 | 1 | WD2(1) |
| | Genbank (KP314034) | Aa-2 | 1 | WD3(1) |
| | Genbank (AF323836) | Aa-3 | 1 | WD4(1) |
| | Genbank (AF323835) | Aa-4 | 1 | WD5(1) |
| | Genbank (MW234506) | Aa-5 | 1 | WD4(1) |
| | Genbank (KP314062) | Aa-6 | 1 | WD6(1) |
| | Genbank (MW234481) | Aa-7 | 1 | WD4(1) |
| | Genbank (MW234478) | Aa-8 | 1 | WD4(1) |
| *A. chinensis* | Lishan,Lishui, Zhejiang | ZT(1~10) | 10 | ZH1(10) |
| | Shakengkou, Lishui, Zhejiang | ZK | 10 | ZH1(9), ZH2(1) |
| | Genbank (KC832305) | Ach-1 | 1 | ZH3(1) |
| | Genbank (KC832307) | Ach-2 | 1 | ZH4(1) |
| | Genbank (MH711099) | Ach-3 | 1 | ZH1(1) |
| | Genbank (MH710876) | Ach-4 | 1 | ZH1(1) |
| | Genbank (KC832316) | Ach-5 | 1 | ZH5(1) |
| | Genbank (KC519784) | Ach-6 | 1 | ZH6(1) |
| | Genbank (KC832302) | Ach-7 | 1 | ZH6(1) |

*(Continued)*

**Table 1.** (Continued)

| Species Name | Locatiy information | Samples No. | Samples size | Haplotype(number) |
|---|---|---|---|---|
| *A. valvata* | Lishan,Lishui, Zhejiang | DT(1~10) | 10 | DE1(10) |
| | Genbank (MG714283) | Av-1 | 1 | DE2(1) |
| | Genbank (MG714281) | Av-2 | 1 | DE3(1) |
| | Genbank (KC519764) | Av-3 | 1 | DE4(1) |
| | Genbank (KC519763) | Av-4 | 1 | DE4(1) |
| | Genbank (KC519762) | Av-5 | 1 | DE4(1) |
| | Genbank (AF323842) | Av-6 | 1 | DE5(1) |
| *A. hemsleyana* | Lishan,Lishui, Zhejiang | CT(1~10) | 10 | CY1(10) |
| | Genbank (MG714284) | Ah-1 | 1 | CY1(1) |
| | Genbank (KC519783) | Ah-2 | 1 | CY2(1) |
| | Genbank (KC519782) | Ah-3 | 1 | CY2(1) |
| | Genbank (KC519781) | Ah-4 | 1 | CY2(1) |
| | Genbank (AF323802) | Ah-5 | 1 | CY3(1) |
| *A. callosa* | Lishan,Lishui, Zhejiang | YT(1~10) | 10 | YS1(10) |
| | Genbank (AF323803) | Aca-1 | 1 | YS2(1) |
| | Genbank (MH808396) | Aca-2 | 1 | YS3(1) |
| | Genbank (MH808395) | Aca-3 | 1 | YS4(1) |
| | Genbank (KC519753) | Aca-4 | 1 | YS5(1) |
| | Genbank (KC519739) | Aca-5 | 1 | YS6(1) |
| | Genbank (AF323804) | Aca-6 | 1 | YS7(1) |
| | Genbank (AF323828) | Aca-7 | 1 | YS8(1) |
| *Saurauia tristyla* | Genbank (KP092594) | ST | 1 | ST1(1) |

Note: *Saurauia tristyla* is the outgroup.

conserved, with 164bp in length. The lengths of ITS1 regions were ranged from 240 to 246 bp with GC content ranging from 49.79% to 60.25%, and the ITS2 regions were ranged from 218 to 223 bp with GC content ranging from 49.77% to 62.73%.

### 3.2 Analysis of haplotype and variable sites

A total of 46 haplotypes were formed in 185 samples (Table 1), among which 9 haplotypes were formed in *A. eriantha*, 5 in *A. polygama*, 4 in *A. melanandra*, 6 in *A. arguta*, 6 in *A. chinensis*, 5 in *A. valvata*, 3 in *A. hemsleyana*, and 5 in *A. callosa*. Analysis of Variation sites in all samples is shown in Table 3. There are 194 variation sites in 8 *Actinidia* species, except for *A. melanandra*, *A. arguta* and *A. chinensis*, the other 5 species have unique Variable sites. For example, bases at 85 bp, 205 bp, 493 bp, 542 bp, 574 bp, 582 bp, and 610 bp of *A. eriantha* are C, T, C, G, C, C, G, respectively. Bases at 28 bp, 79 bp, 83 bp, 109 bp, 128 bp, 448 bp, and 501 bp of *A. hemsleyana* are T, A, A, T, A, T, T, respectively. These mutated loci can be considered as the unique identification loci for the root of *A. eriantha*, *A. polygama*, *A. valvata*, *A. hemsleyana* and *A. callosa*.

### 3.3 K2P genetic distance analysis

The genetic distance of 8 *Actinidia* species based on ITS sequence is shown in Tables 4 and 5. The intra-specific genetic distance of *A. callosa*, *A. arguta* and *A. melanandra* is 2.18%, 2.13% and 2.26%, respectively. And then, the intra-specific genetic distance of *A. eriantha* and *A.*

**Table 2. Composition of ITS sequences of 8 *Actinidia* species.**

| Species Name | Haplotype | Sequences lengthens of ITS1 (bp) | GC content of ITS1 (%) | Sequences lengthens of 5.8S (bp) | G+C % | Sequences lengthens of ITS2 (bp) | G+C content of ITS2 (%) | Sequences lengthens of ITS (bp) |
|---|---|---|---|---|---|---|---|---|
| *A. eriantha* | MH1 | 245 | 55.51 | 164 | 53.66 | 220 | 56.82 | 629 |
| *A. eriantha* | MH2 | 244 | 55.33 | 164 | 53.66 | 220 | 56.82 | 628 |
| *A. eriantha* | MH3 | 245 | 55.92 | 164 | 53.66 | 220 | 56.82 | 629 |
| *A. eriantha* | MH4 | 245 | 55.92 | 164 | 53.66 | 220 | 57.27 | 629 |
| *A. eriantha* | MH5 | 245 | 55.51 | 164 | 53.66 | 220 | 56.82 | 629 |
| *A. eriantha* | MH6 | 245 | 55.10 | 164 | 53.66 | 220 | 56.82 | 629 |
| *A. eriantha* | MH7 | 245 | 56.73 | 164 | 53.66 | 220 | 57.27 | 629 |
| *A. eriantha* | MH8 | 245 | 56.33 | 164 | 53.66 | 220 | 56.82 | 629 |
| *A. eriantha* | MH9 | 245 | 56.33 | 164 | 53.66 | 220 | 56.36 | 629 |
| *A. polygama* | GZ1 | 244 | 51.64 | 164 | 52.44 | 219 | 55.71 | 627 |
| *A. polygama* | GZ2 | 243 | 51.85 | 164 | 53.05 | 219 | 54.79 | 626 |
| *A. polygama* | GZ3 | 243 | 51.44 | 164 | 53.05 | 219 | 54.79 | 626 |
| *A. polygama* | GZ4 | 244 | 51.23 | 164 | 52.44 | 219 | 55.71 | 627 |
| *A. polygama* | GZ5 | 242 | 51.65 | 164 | 53.05 | 219 | 54.79 | 625 |
| *A. melanandra* | HL1 | 244 | 59.84 | 164 | 54.27 | 220 | 62.73 | 628 |
| *A. melanandra* | HL2 | 246 | 58.13 | 168 | 52.98 | 222 | 61.26 | 636 |
| *A. melanandra* | HL3 | 245 | 59.18 | 164 | 54.27 | 220 | 62.73 | 629 |
| *A. melanandra* | HL4 | 245 | 58.78 | 164 | 54.27 | 220 | 60.45 | 629 |
| *A. arguta* | WD1 | 245 | 58.78 | 164 | 54.27 | 220 | 61.82 | 629 |
| *A. arguta* | WD2 | 244 | 59.84 | 164 | 54.27 | 220 | 62.73 | 628 |
| *A. arguta* | WD3 | 244 | 58.61 | 164 | 54.88 | 220 | 62.73 | 628 |
| *A. arguta* | WD4 | 244 | 60.25 | 164 | 54.27 | 220 | 62.73 | 628 |
| *A. arguta* | WD5 | 240 | 59.17 | 164 | 54.27 | 220 | 62.73 | 624 |
| *A. arguta* | WD6 | 243 | 58.02 | 164 | 54.88 | 220 | 62.73 | 627 |
| *A. chinensis* | ZH1 | 243 | 49.79 | 164 | 54.27 | 219 | 49.77 | 626 |
| *A. chinensis* | ZH2 | 243 | 50.62 | 164 | 54.27 | 219 | 51.60 | 626 |
| *A. chinensis* | ZH3 | 243 | 50.62 | 164 | 54.27 | 219 | 51.60 | 626 |
| *A. chinensis* | ZH4 | 243 | 50.21 | 164 | 54.27 | 219 | 52.05 | 626 |
| *A. chinensis* | ZH5 | 243 | 50.62 | 164 | 54.27 | 219 | 52.05 | 626 |
| *A. chinensis* | ZH6 | 243 | 50.62 | 164 | 54.27 | 219 | 52.05 | 626 |
| *A. valvata* | DE1 | 244 | 55.74 | 164 | 53.66 | 221 | 61.09 | 629 |
| *A. valvata* | DE2 | 244 | 55.74 | 164 | 53.66 | 221 | 60.63 | 629 |
| *A. valvata* | DE3 | 243 | 55.56 | 164 | 53.66 | 221 | 61.09 | 628 |
| *A. valvata* | DE4 | 244 | 54.92 | 164 | 53.66 | 223 | 60.99 | 631 |
| *A. valvata* | DE5 | 244 | 55.74 | 164 | 53.66 | 221 | 60.63 | 629 |
| *A. hemsleyana* | CY1 | 243 | 50.21 | 164 | 53.05 | 219 | 50.23 | 626 |
| *A. hemsleyana* | CY2 | 244 | 50.00 | 164 | 53.05 | 219 | 50.68 | 627 |
| *A. hemsleyana* | CY3 | 243 | 50.21 | 164 | 53.66 | 218 | 50.46 | 625 |
| *A. callosa* | YS1 | 244 | 56.15 | 164 | 54.27 | 219 | 56.16 | 627 |
| *A. callosa* | YS2 | 243 | 55.97 | 164 | 54.27 | 219 | 56.62 | 626 |
| *A. callosa* | YS3 | 244 | 55.33 | 164 | 54.27 | 219 | 56.62 | 627 |
| *A. callosa* | YS4 | 244 | 58.20 | 164 | 54.27 | 220 | 57.27 | 628 |
| *A. callosa* | YS5 | 244 | 58.20 | 164 | 54.27 | 221 | 57.01 | 629 |
| *A. callosa* | YS6 | 244 | 56.15 | 164 | 54.27 | 219 | 56.62 | 627 |
| *A. callosa* | YS7 | 242 | 56.61 | 164 | 54.27 | 218 | 56.42 | 624 |
| *A. callosa* | YS8 | 243 | 53.09 | 164 | 54.27 | 220 | 56.82 | 627 |

**Table 3. Analysis of base variation in 8 species of *Actinidia*.**

| Haplotype | ITS1 28 | 48 | 73 | 79 | 83 | 85 | 92 | 93 | 109 | 115 | 122 | 128 | 147 | 184 | 205 | 232 | 5.8S 346 | 352 | 361 | 416 | 422 | 435 | 446 | 436 | 455 | 487 | 493 | 501 | 530 | ITS2 542 | 548 | 602 | 603 | 604 | 605 | 606 | 610 | 620 | 640 | GenbanK No. |
|---|---|---|---|---|---|---|---|---|---|---|---|---|---|---|---|---|---|---|---|---|---|---|---|---|---|---|---|---|---|---|---|---|---|---|---|---|---|---|---|---|
| MH1 | C | C | C | T | T | C | C | C | C | — | G | — | G | T | T | C | C | A | T | G | T | A | C | A | — | C | C | C | A | G | T | T | T | A | C | T | C | G | T | / | OK036712 |
| MH2 | * | * | * | * | * | * | * | * | * | * | * | * | * | * | * | * | * | * | * | * | * | * | * | * | * | * | * | * | * | * | * | * | * | * | * | * | * | * | G | / | OK036713 |
| MH3 | * | * | * | * | * | * | * | * | * | / | * | / | * | * | * | * | * | * | * | * | * | * | * | * | * | * | * | * | * | * | * | * | * | * | * | * | * | * | * | / | OK036795 |
| MH4 | * | * | * | * | * | * | * | * | * | / | * | / | * | * | * | * | * | * | * | * | * | * | * | * | * | * | * | * | * | * | * | * | * | * | * | * | * | * | * | / | OK036796 |
| MH5 | * | * | * | * | * | * | * | * | * | / | * | / | * | * | * | * | * | * | * | * | * | * | * | * | * | * | * | * | * | * | * | * | * | * | * | * | * | * | * | / | OK036797 |
| MH6 | * | * | * | * | * | * | * | * | * | / | * | / | * | * | * | * | * | * | * | * | * | * | * | * | * | * | * | * | * | * | * | * | * | * | * | * | * | * | * | / | MG714287 |
| MH7 | * | * | * | * | * | * | * | * | * | / | * | / | * | * | * | * | * | * | * | * | * | * | * | * | * | * | * | * | * | * | * | * | * | * | * | * | * | * | * | / | KP314044 |
| MH8 | * | * | * | * | * | * | * | * | * | / | * | / | * | * | * | * | * | * | * | * | * | * | * | * | * | * | * | * | * | * | * | * | * | * | * | * | * | * | * | / | KP314039 |
| MH9 | * | * | * | * | * | * | * | * | * | / | * | / | * | * | * | * | * | * | * | * | * | * | * | * | * | * | * | * | * | * | * | * | * | * | * | * | * | * | * | / | KP314038 |
| CY1 | T | * | A | A | A | A | T | T | T | A | * | A | * | * | A | A | A | * | * | * | * | * | * | * | * | * | T | T | * | A | G | * | * | * | * | * | * | T | * | / | OK036804 |
| CY2 | T | * | A | A | A | A | T | T | T | A | * | A | * | * | A | A | A | * | * | * | * | * | * | * | * | * | T | T | * | A | G | * | * | * | * | * | * | T | * | / | KC519783 |
| CY3 | T | * | A | A | A | A | T | T | T | A | * | A | * | * | A | A | A | * | * | * | * | * | * | * | * | * | T | T | * | A | G | * | * | * | * | * | * | T | * | / | AF323802 |
| DE1 | * | G | A | * | * | * | T | C | C | / | C | / | C | * | A | A | A | A | C | G | C | T | C | * | G | * | C | T | G | A | G | A | C | T | C | G | G | C | C | T | OK036803 |
| DE2 | * | G | A | * | * | * | T | C | C | / | C | / | C | * | A | A | A | A | C | G | C | T | C | * | G | * | C | T | G | A | G | A | C | T | C | G | G | C | C | T | MG714283 |
| DE3 | * | G | A | * | * | * | T | C | C | / | C | / | C | * | A | A | A | A | C | G | C | T | C | * | G | * | C | T | G | A | G | A | C | T | C | G | G | C | C | T | MG714281 |
| DE4 | * | G | A | * | * | * | T | C | C | / | C | / | C | * | A | A | A | A | C | G | C | T | C | * | G | * | C | T | G | A | G | A | C | T | C | G | G | C | C | T | KC519764 |
| DE5 | * | G | A | * | * | * | T | C | C | / | C | / | C | * | A | A | A | A | C | G | C | T | C | * | G | * | C | T | G | A | G | A | C | T | C | G | G | C | C | T | AF323842 |
| GZ1 | * | T | T | * | * | * | * | * | * | * | * | * | T | * | A | A | A | * | * | A | * | A | T | A | * | T | * | T | * | A | G | * | G | C | T | G | C | T | * | / | OK036798 |
| GZ2 | * | T | T | * | * | * | * | * | * | * | * | * | T | G | A | A | A | * | * | A | * | A | T | A | * | T | * | T | * | A | G | * | G | T | T | T | C | T | T | / | KP314060 |
| GZ3 | * | T | T | * | * | G | * | * | * | * | * | * | T | G | A | A | A | * | * | A | * | A | T | A | * | T | * | T | * | A | G | * | G | C | T | G | C | T | * | / | KP314051 |
| GZ4 | * | T | T | * | * | * | * | * | * | * | * | * | T | * | A | A | A | * | * | A | * | A | T | A | * | T | * | T | * | A | G | * | G | T | T | T | C | T | T | / | KC519766 |
| GZ5 | * | T | T | * | * | G | * | * | * | * | * | * | T | G | A | A | A | * | * | A | * | A | T | A | * | T | * | T | * | A | G | * | G | C | T | G | C | T | * | / | AF323796 |
| HL1 | * | * | G | * | * | * | * | * | * | * | * | * | * | * | A | A | A | * | * | * | * | G | * | * | * | * | * | T | * | A | G | * | * | G | T | * | C | C | * | / | OK036799 |
| HL2 | * | * | T | * | * | * | * | * | * | * | * | * | * | G | A | A | A | * | * | * | * | * | * | * | * | * | * | T | * | A | G | * | * | T | T | * | C | C | * | / | AF443211 |
| HL3 | * | * | G | * | * | * | * | * | * | * | * | * | * | * | A | A | A | * | * | * | * | G | * | * | * | * | * | T | * | A | G | * | * | G | T | * | C | C | * | / | MG714282 |
| HL4 | * | / | / | * | * | * | * | * | * | * | * | * | * | * | A | A | A | * | * | * | * | * | * | * | * | * | * | T | * | A | G | * | * | T | T | * | C | C | * | / | AF323808 |
| WD1 | * | T | T | G | * | * | * | * | * | * | * | * | * | * | A | A | A | * | * | * | * | * | * | * | * | * | * | T | * | A | G | * | * | G | T | * | C | C | * | / | OK036800 |
| WD2 | * | / | / | * | * | * | * | * | * | * | * | * | * | * | A | A | A | * | * | * | * | * | * | * | * | * | * | T | * | A | G | * | * | G | T | * | C | C | * | / | KP314061 |
| WD3 | * | T | T | G | * | * | * | * | * | * | * | * | * | * | A | A | A | * | * | * | * | * | * | * | * | * | * | T | * | A | G | * | * | G | T | * | C | C | * | / | KP314034 |
| WD4 | * | / | / | G | * | * | * | * | * | * | * | * | * | * | A | A | A | * | * | * | * | * | * | * | * | * | * | T | * | A | G | * | * | G | T | * | C | C | * | / | AF323836 |
| WD5 | * | / | / | G | * | * | * | * | * | * | * | * | * | * | A | A | A | * | * | * | * | * | * | * | * | * | * | T | * | A | G | * | * | G | T | * | C | C | * | / | AF323835 |
| WD6 | * | T | T | * | * | * | * | * | * | * | * | * | * | * | A | A | A | * | * | * | * | * | * | * | * | * | * | T | * | A | G | * | * | G | T | * | C | C | * | / | KP314062 |
| YS1 | * | * | * | * | * | * | * | * | * | * | * | * | * | * | A | A | A | * | * | * | C | * | C | * | * | G | * | T | * | A | G | * | * | G | T | * | T | T | * | / | OK036805 |
| YS2 | * | * | * | * | * | * | * | * | * | * | * | * | * | * | A | A | A | * | * | * | C | * | C | * | * | G | * | T | * | A | G | * | * | G | T | * | T | T | * | / | AF323803 |
| YS3 | * | * | * | * | * | * | * | * | * | * | * | * | * | * | A | A | A | * | * | * | C | * | C | * | * | G | * | T | * | A | G | T | * | G | T | * | T | * | * | / | MH808396 |
| YS4 | * | * | * | * | * | * | * | * | * | * | * | * | * | * | A | A | A | * | * | * | C | * | C | * | * | G | * | T | * | A | T | * | * | G | T | * | C | * | * | / | MH808395 |

(*Continued*)

**Table 3.** (Continued)

| Haplotype | base variation | | | | | | | | | | | | | | | | | | | | | | | | | | | | | | | | | | GenbanK No. |
|---|---|---|---|---|---|---|---|---|---|---|---|---|---|---|---|---|---|---|---|---|---|---|---|---|---|---|---|---|---|---|---|---|---|---|---|
| | ITS1 | | | | | | | | | | | | | 5.8S | | | | | | | | | | ITS2 | | | | | | | | | | | |
| | 248 | 273 | 79 | 83 | 85 | 92 | 115 | 122 | 128 | 147 | 184 | 205 | 232 | 416 | 422 | 435 | 436 | 446 | 448 | 487 | 493 | 501 | 530 | 542 | 574 | 582 | 600 | 603 | 604 | 605 | 606 | 610 | 620 | 664 | |
| YS5 | * | * | * | * | T | * | * | * | / | * | * | A | * | * | C | * | * | G | * | * | T | * | * | A | T | T | * | * | * | * | * | C | * | / | KC519753 |
| YS6 | * | * | * | * | T | * | * | * | / | * | * | A | * | * | C | * | * | G | * | * | T | * | * | A | T | G | * | * | * | * | * | T | * | / | KC519739 |
| YS7 | * | * | * | * | T | * | * | * | / | * | * | A | * | * | C | * | * | G | * | * | T | * | * | A | T | G | * | * | * | * | * | T | * | / | AF323804 |
| YS8 | * | * | * | C | T | * | * | * | / | * | * | A | * | * | C | * | * | G | * | * | T | * | * | A | T | T | * | * | * | * | * | C | * | / | AF323828 |
| ZH1 | * | * | T | C | T | * | * | * | / | A | * | A | * | * | * | * | * | * | * | * | T | * | * | A | T | G | * | * | * | * | * | T | * | / | OK036801 |
| ZH2 | * | * | T | C | T | * | * | * | / | * | * | A | * | * | * | * | * | * | * | * | T | * | * | A | T | G | * | * | * | * | * | T | * | / | OK036802 |
| ZH3 | * | * | T | * | T | * | * | * | / | A | * | A | * | * | * | * | * | * | * | * | T | * | * | A | T | G | * | * | * | * | * | T | * | / | KC832305 |
| ZH4 | * | * | T | C | T | * | * | * | / | * | * | A | * | * | * | * | * | * | * | * | T | * | * | A | T | G | * | * | * | * | * | T | * | / | KC832307 |
| ZH5 | * | * | T | * | T | * | * | * | / | * | * | A | * | * | * | * | * | * | * | * | T | * | * | A | T | G | * | * | * | * | * | T | * | / | KC832316 |
| ZH6 | * | * | T | * | T | * | * | * | / | A | * | A | * | * | * | * | * | * | * | * | T | * | * | A | T | G | * | * | * | * | * | T | * | / | KC519784 |

Note: M = A+C, R = A+G, /, base deletion.

**Table 4. Intra-specific genetic distance of 8 species of *Actinidia*.**

| Species name | YS | WD | ZH | HL | GZ | DE | CY | MH |
|---|---|---|---|---|---|---|---|---|
| Genetic distance | 2.18% | 2.13% | 1.65% | 2.26% | 0.94% | 0.35% | 0.59% | 0.48% |

Note: DE = *A. valvata*, GZ = *A. polygama*, HL = *A. melanandra*h, CY = *A. hemsleyana*, MH = *A. eriantha*, ZH = *A. chinensis*, YS = *A. callosa*, WD = *A. arguta*.

*valvata* is less than *A. callosa*, *A. arguta* and *A. melanandra*, which is 0.48% and 0.35%, respectively.

The inter-specific genetic distance among 8 *Actinidia* species ranged from 2.28% to 11.00%. The genetic distance between *A. melanandra* and *A. hemsleyana* was the highest (11.00%), while the genetic distance between *A. melanandra* and *A. arguta* was the lowest (2.28%). For *A. eriantha*, there was a largest genetic distance between *A. eriantha* and *A. arguta* (9.56%), and the smallest genetic distance between *A. eriantha* and *A. callosa* (4.77%).

## 3.4 Phylogenetic analysis

ITS1, ITS2 and ITS sequences were tested 1000 times by bootstrap method to build ML phylogenetic tree (Fig 2), with *Saurauia tristyla* as the outgroup. Phylogenetic tree of ITS sequences showed that all of the tested samples can be grouped into two main clusters. One cluster includes *A. eriantha*, *A. callosa*, *A. chinensis* and *A. hemsleyana*, and each species also clustered separate group respectively. The other cluster includes *A. polygama*, *A. valvata*, *A. arguta* and *A. melanandrah*, among which *A. polygama* and *A. valvata* clustered into each separate group respectively, and *A. arguta* and *A. melanandrah* clustered into one group. Phylogenetic tree of ITS1 sequences showed that *A. arguta* and *A. melanandrah* clustered into one group, and the other 6 species were group into another category. Moreover, in this category, *A. eriantha*, *A. hemsleyana*, *A. polygama* and *A. valvata* were separately distinguished, and *A. callosa* and *A. chinensis* scattered in the category. Phylogenetic tree of ITS2 sequences shows similar results with ITS sequence, except *A. hemsleyana* and *A. chinensis* which are clustered into one group. For ITS sequences, except *A. arguta* and *A. melanandrah* the other 6 species can be distinguished separately from others. For ITS1 sequences, *A. eriantha*, *A. hemsleyana*, *A. polygama* and *A. valvata* can be distinguished separately from the other 4 varieties. For ITS2 sequences, *A. eriantha*, *A. callosa*, *A. polygama* and *A. valvata* can be distinguished separately from the other 4 varieties.

**Table 5. Genetic distances among 8 species of *Actinidia*.**

| Species name | YS | WD | ZH | HL | GZ | DE | CY | MH |
|---|---|---|---|---|---|---|---|---|
| YS | | | | | | | | |
| WD | 9.86% | | | | | | | |
| ZH | 5.69% | 10.15% | | | | | | |
| HL | 10.11% | 2.28% | 10.28% | | | | | |
| GZ | 6.95% | 9.73% | 6.42% | 9.94% | | | | |
| DE | 9.01% | 9.72% | 9.60% | 10.04% | 7.51% | | | |
| CY | 5.99% | 10.89% | 2.88% | 11.00% | 6.53% | 10.46% | | |
| MH | 4.77% | 9.56% | 6.19% | 9.87% | 5.97% | 7.77% | 6.36% | |

Note: DE = *A. valvata*, GZ = *A. polygama*, HL = *A. melanandra*h, CY = *A. hemsleyana*, MH = *A. eriantha*, ZH = *A. chinensis*, YS = *A. callosa*, WD = *A. arguta*.

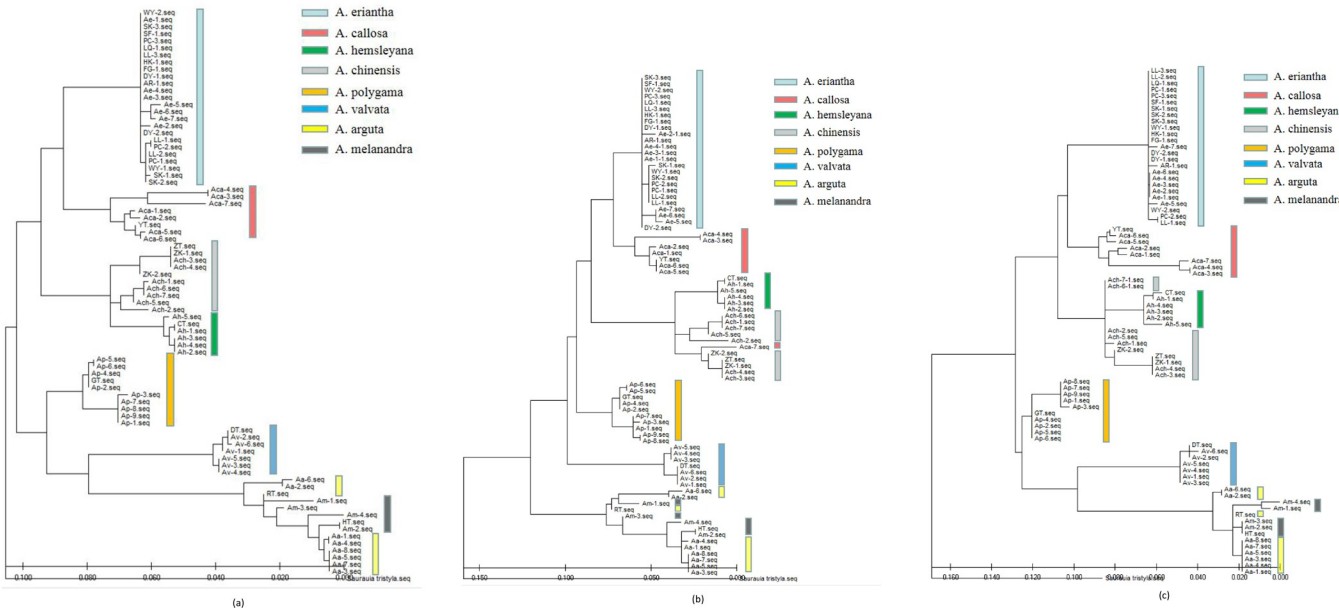

**Fig 2. Phylogenetic tree of 8 species of *Actinidia*.** (a) Analysis of ITS sequences fragments; (b) Analysis of ITS1 sequences fragments; (c) Analysis of ITS2 sequences fragments.

## 4 Discussion

As early as in the "compendium of materia medica", there is a record of the medicinal plants of the *Actinidia*. Many *Actinidia* species with high commercial value have received considerable attention because of their edible fruits, medical value and ornamental values [14–16]. Therefore, their accurate identification is highly in demand. However, currently the identification of *Actinidia* mainly depends on the characteristics of the original plant, which requires a high degree of experience, especially when the plant is dry [17]. For example, *A. eriantha* and *A. chinensis* have a high degree of similarity, in terms of the body of the plant, the size of the fruit and the coat characteristics of the fruit, which are very closely. The only distinguishing feature is that the coat of *A. eriantha* is milky white, thick and dense [18]. The ITS region has been used as a DNA barcode to authenticate various plants with similar morphological traits [19]. In this study, the ITS sequences were used for barcoding *Actinidia* species for the first time.

The ITS sequence analysis of 8 *Actinidia* species showed that there were 194 mutants, including *A. eriantha* (85 bp C、205 bp T、493 bp C、542 bp G、574 bp C、582 bp T、610 bp G), *A. hemsleyana* (28 bp T、79 bp A、83 bp A、87 bp C、109 bp T、128 bp A、448 bp T、501 bp T), *A. valvata* (48bpG、73bpA、92bpT、95bpC、122bpC、184bpC、232bpA、435bpC、487bpC、530bpG、602bpC、620bpT、640bpT), *A. polygama* (93 bp T、115 bp A、147 bp T、416 bp A、436 bp T、603 bp G、605 bp T、606 bp C)and *A. callosa* (422 bp C、446 bp G). All of them have their own specific loci. These specific loci can be used as the basis for the identification of several *Actinidia* species.

Many researchers have reported that ITS region has the advantages of high intra-specific conservation, large inter-specific variation, small fragment, easy amplification, easy analysis and high success rate for species identification [20, 21]. Similar results were found in our study. We found that the genetic distance between the most samples from the same species in this study is less than 1% (*A. eriantha*, *A. hemsleyana*, *A. valvata*, *A. polygama*), and there were

no specific loci differentiating among them. This feature is thus useful for identifying different *Actinidia* species and related species.

The phylogenetic tree which was constructed with ITS, ITS1 and ITS2 region showed that the ITS sequences have higher identification ability in 8 *Actinidia* species. However, these three barcodes have the same identification ability for *A. eriantha*, *A. hemsleyana* and *A. valvata*. The ITS regions indicated that different samples from same species can be grouped together, except for *A. arguta* and *A. melanandrah*. Previoues study reported two groups, one group includes *A. eriantha*, *A. callosa*, *A. chinensis* and *A. hemsleyana*, the other group includes *A. polygama*, *A. valvata*, *A. arguta* and *A. melanandrah*, which are based on *mat*K gene. Whereas, *A. eriantha*, *A. callosa*, *A. chinensis*, *A. hemsleyana* and *A. polygama* clustered into one group based on *rbc*L gene. Our study is consistent with the results indicated by *mat*K gene. Therefore, the ITS sequences are useful for species identification, as well as contribute to the phylogenetic analysis of *Actinidia* and its related species.

## 5 Conclusion

Our study demonstrates that the ITS sequences possess high species discriminability and it could be a useful DNA barcode for *Actinidia* species. However, more *Actinidia* species should be collected in the future to verify whether ITS sequences that could be used to identify all species of *Actinidia*. In addition, ML tree analyses provided a solid evidence that the ITS sequences have the potential in the phylogenetic analysis of *Actinidia* and its related species.

## Supporting information

**S1 Table. Mutation loci analysis of 8 *Actinidia*.**
(XLS)

## Acknowledgments

Many thanks to Wei Ding and Shuang Liu for their help in the research process, and thanks to Weiwen Qiu for his guidance.

## Author Contributions

**Conceptualization:** Houxing Lei.

**Data curation:** Yixing Zhang.

**Formal analysis:** Xiaoqin Zhang, Jiale Mao.

**Funding acquisition:** Xiaoqin Zhang.

**Methodology:** Xiaoqin Zhang.

**Resources:** Zunjing Zhang.

**Software:** Yan Lan.

**Writing – original draft:** Xiaoqin Zhang.

**Writing – review & editing:** Zunjing Zhang.

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
