## [Decision Letter · Decision Letter 0]

6 Jun 2022

PONE-D-22-07942The ITS Analysis and Identification of Actinidia eriantha and its Related SpeciesPLOS ONE

Dear Dr. Mao,

Thank you for submitting your manuscript to PLOS ONE. After careful consideration, we feel that it has merit but does not fully meet PLOS ONE’s publication criteria as it currently stands. Therefore, we invite you to submit a revised version of the manuscript that addresses the points raised during the review process.

We look forward to receiving your revised manuscript.

Kind regards,

Evangelia V. Avramidou, PhD

Academic Editor

PLOS ONE

Journal Requirements:

3. Please include your tables as part of your main manuscript and remove the individual files. Please note that supplementary tables (should remain/ be uploaded) as separate "supporting information" files.

4. We suggest you thoroughly copyedit your manuscript for language usage, spelling, and grammar. If you do not know anyone who can help you do this, you may wish to consider employing a professional scientific editing service.

Whilst you may use any professional scientific editing service of your choice, PLOS has partnered with both American Journal Experts (AJE) and Editage to provide discounted services to PLOS authors. Both organizations have experience helping authors meet PLOS guidelines and can provide language editing, translation, manuscript formatting, and figure formatting to ensure your manuscript meets our submission guidelines. To take advantage of our partnership with AJE, visit the AJE website (http://aje.com/go/plos) for a 15% discount off AJE services. To take advantage of our partnership with Editage, visit the Editage website (www.editage.com) and enter referral code PLOSEDIT for a 15% discount off Editage services.  If the PLOS editorial team finds any language issues in text that either AJE or Editage has edited, the service provider will re-edit the text for free.

The name of the colleague or the details of the professional service that edited your manuscriptA copy of your manuscript showing your changes by either highlighting them or using track changes (uploaded as a *supporting information* file)A clean copy of the edited manuscript (uploaded as the new *manuscript* file)”"

6. Thank you for stating the following financial disclosure:

“Ms. XZ was funded by [Zhejiang Provincial Natural Science Foundation of China] grant number [LGF20H280005], Mr. HL was funded by[development project in Lishui city] grant number [2020ZDYF 15] and Ms. YL was funded by [Lishui Science and Technology project] grant number [2021SJZC039].

YES - Specify the role(s) played.”

Additional Editor Comments:

Dear authors,

based on reviewer's comments and also from my expertise your article has many deficiencies and needs further improvement. So please answer reviewers's somments and additionaly please answer to the following comments:

1. why did you prefer ITS although discriminations power of SSR are more accurate? is it less expensive to do a sequencing analysis rather genotyping?

2. please provide some photos also for the difficult part of morphological description

3. finally the manuscript needs to be critically proof read before resubmitted.

With kind regards

Reviewers' comments:

Reviewer's Responses to Questions

**Comments to the Author**

1. Is the manuscript technically sound, and do the data support the conclusions?

Reviewer #1: Partly

2. Has the statistical analysis been performed appropriately and rigorously? 

Reviewer #1: I Don't Know

3. Have the authors made all data underlying the findings in their manuscript fully available?

Reviewer #1: No

4. Is the manuscript presented in an intelligible fashion and written in standard English?

Reviewer #1: No

5. Review Comments to the Author

Reviewer #1: 1. The authors suggested ITS as a reliable marker for species identification of Actinidia eriantha and its Related Species (n=185 samples which includes 53 samples from Genbank). It is not clear from the methodology whether these are their own samples or from others, thus questioning the reliability of species vouchering. This is crucial to be clarified due to: “However, it is difficult to identify them due to the diverse variability in morphological characters[4].”, as stated by the authors. And it's also not clear if the new barcodes generated from the other haplotypes listed in Table 1 have been submitted and verified by Genbank.

2. Furthermore, the methods lack details and are only vaguely described, without proper citation. Eg: "Fresh young leaves of the collected samples were used for genomic DNA isolation, as previously reported". A supplementary data of the characteristics of all 8 species would be extremely helpful to convince the readers on the morphological species identification.

3. The authors generated a good dataset but the way it was reported and discussed is really confusing and only partly supports the overall conclusion.

4. Overall, the manuscript is poorly written and difficult to follow. The manuscript needs to be critically proof read before resubmitted for further review. Obvious language mistakes are observed throughout the manuscript.

6. PLOS authors have the option to publish the peer review history of their article (what does this mean?). If published, this will include your full peer review and any attached files.

Reviewer #1: No

---

## [Author Response · Author response to Decision Letter 0]

4 Aug 2022

1. Dr. Bin Wu of Zhejiang University thoroughly copyedit manuscript for language usage, spelling, and grammar.

2. The minimal data set underlying the results described in your manuscript can be found in Genbank(accession numbers see S1 Table).

3. Ms. Xiaoqin Zhang was funded by [Zhejiang Provincial Natural Science Foundation of China] grant number [LGF20H280005], Mr. Houxing Lei was funded by[development project in Lishui city] grant number [2020ZDYF 15] and Ms. Yan Lan was funded by [Lishui Science and Technology project] grant number [2021SJZC039].

Ms. Xiaoqin Zhang – Methodology and Writing–original draft

Mr. Houxing Lei –Conceptualization

Ms. Yan Lan –Software

4. We add Leaf image of 8 species of Actinidia in manuscript for the difficult part of morphological description.

5. The research material is plant leaves , it is not involve ethical protection and genetic information protection, so no additional information regarding the permits for the work we need.

6. SSR has some shortcomings: 1) the detection of SSR polymorphism largely depends on the effect of PCR amplification; 2) The existing SSR markers are limited in number, and can not mark all functional genes; 3) The development and synthesis of primers are expensive, difficult and time-consuming.

---

## [Decision Letter · Decision Letter 1]

26 Aug 2022

The ITS Analysis and Identification of Actinidia eriantha and its Related Species

PONE-D-22-07942R1

Dear Dr. Mao,

We’re pleased to inform you that your manuscript has been judged scientifically suitable for publication and will be formally accepted for publication once it meets all outstanding technical requirements.

Kind regards,

Evangelia V. Avramidou, PhD

Academic Editor

PLOS ONE

Additional Editor Comments (optional):

Dear authors,

based on the answers you provided the manuscript is now suitable for publication.

Reviewers' comments:

Reviewer's Responses to Questions

**Comments to the Author**

1. If the authors have adequately addressed your comments raised in a previous round of review and you feel that this manuscript is now acceptable for publication, you may indicate that here to bypass the “Comments to the Author” section, enter your conflict of interest statement in the “Confidential to Editor” section, and submit your "Accept" recommendation.

Reviewer #1: (No Response)

2. Is the manuscript technically sound, and do the data support the conclusions?

Reviewer #1: Partly

3. Has the statistical analysis been performed appropriately and rigorously? 

Reviewer #1: No

4. Have the authors made all data underlying the findings in their manuscript fully available?

Reviewer #1: Yes

5. Is the manuscript presented in an intelligible fashion and written in standard English?

Reviewer #1: No

6. Review Comments to the Author

Reviewer #1: Aside from the addition of leaf images (which also lacks proper citation), the manuscript was barely improved from the first version.

7. PLOS authors have the option to publish the peer review history of their article (what does this mean?). If published, this will include your full peer review and any attached files.

Reviewer #1: No

---

## [Editor Report · Acceptance letter]

7 Sep 2022

PONE-D-22-07942R1 

The ITS Analysis and Identification of *Actinidia eriantha* and its Related Species 

Dear Dr. Mao:

I'm pleased to inform you that your manuscript has been deemed suitable for publication in PLOS ONE. Congratulations! Your manuscript is now with our production department. 

Kind regards, 

on behalf of

Dr. Evangelia V. Avramidou 

Academic Editor

PLOS ONE